# Development and Properties of Recycled Biomass Fly Ashes Modified Mortars

**Julien Hubert [1]**, **Sophie Grigoletto [1]**, **Frédéric Michel [1]**, **Zengfeng Zhao [2]** and **Luc Courard [1,*]**

1   Urban and Environmental Engineering, Building Materials, University of Liège, 4000 Liège, Belgium; julien.hubert@uliege.be (J.H.); sophie.grigoletto@uliege.be (S.G.); frederic.michel@uliege.be (F.M.)
2   Department of Structural Engineering, College of Civil Engineering, Tongji University, Shanghai 200092, China; zengfengzhao@tongji.edu.cn
*   Correspondence: luc.courard@uliege.be

**Abstract:** The production of biomass fly ash has been increasing every year in Europe, reaching 5.5 million tons in 2020. Fly ash produced by burning biomass is not yet accepted in the standards as a substitute material for cement in mortar and concrete. In a first approach, the substitution limit of biomass ash is determined by comparing the mechanical strengths (among others, compressive strength), fresh state properties and hardened properties of mortars produced with fly ash with those of mortars produced with coal fly ash (EN 450-1 and ASTM C618). Masonry and rendering mortars have been designed with different substitution rates of fly ashes from wood combustion in thermal power plants. Although there is an overall decrease in performance, mortars made with biomass ash retain properties that make them suitable for use in masonry (loss of 13% compressive strength for masonry mortars with 10% substitution rate after 90 days) or rendering (loss of 20% compressive strength for rendering mortars with 10% substitution rate after 90 days). Water absorption and porosity (24.1 and 23.7% for masonry and rendering mortars, respectively) are, however, not significantly modified, which potentially contributes to good durability properties.

**Keywords:** biomass ashes; cement; masonry mortar; rendering mortar; design; properties





## 1. Introduction

Concrete is the most widely used material for infrastructure development with a global yearly consumption approaching 30 billion tons [1]. The production of cement releases approximately 8–9% of the global anthropogenic $CO_2$ emissions. To respect the targets of the 2015 Paris Agreement on climate change, reducing the amount of $CO_2$ generated by cement production is crucial but remains a challenge. One of the strategies to achieve that goal consists of replacing part of the Portland cement in the concrete composition with an industrial by-product. A classical supplementary cementitious material (SCM) is coal fly ash (FA) [2]. FA is obtained by recovering the particles in the flue gas stacks after the combustion of coal. FA is very efficient and largely used by cement manufacturers because it contains silica and aluminum oxides [3]. The specifications and conformity criteria for FA are governed by EN 450-1 and ASTM C618 [4]. The manufacture of cements containing fly ash is covered by EN 197-1. Fly ash used in mortars and concretes must fulfill a number of physical and chemical requirements such as:

- Loss on ignition (EN 196-2): it must not exceed 5% by mass in order to limit the amount of unburnt carbon in the fly ash;
- Chloride content $Cl^-$ (EN 196-21): not to exceed 0.1% by mass;
- Sulphuric anhydride content $SO_3$ (EN 196-2): must not exceed 3% by mass;
- Free calcium oxide content (EN 451-1): must not exceed 1% by mass;
- Fineness (EN 451-2): the maximum value of the fineness must not exceed 40%;
- Activity index (EN 196-1): the activity index at 28 days and 90 days must be greater than 75% and 85%, respectively.

The use of FA in the manufacture of masonry and plastering mortars as a partial replacement for Portland cement or blended cement is highly recommended for rheological reasons [5]. Fairly coarse FA is used to make mortars suitable for masonry and plastering joints. It is easily possible to produce mortars with 20 to 40% of cement replacement by FA with sufficient strength [6]. From a technical point of view, the use of FA in mortars/concretes can improve mechanical strength, physical and chemical ageing resistance and fresh flow properties [7]. The hydration of masonry cement into which FA is incorporated always releases $Ca(OH)_2$ which reacts with the FA to form constituents such as hydrated calcium silicate which contributes to increased mechanical strength.

However, in some countries, ASTM compliant FA is hard to come by and other new sources of supplementary cementitious materials are needed.

Fly ash from biomass (BA) could prove to be an effective alternative to classical FA and has garnered more and more attention in recent years. BA is the residue, mainly alkaline, of the combustion and the incineration of various organic and mineral plant materials, both natural and non-fossil (wood, plants) [8]. Biomass is usually composed of wood waste from forestry, farm waste or waste from the food industry. In 2009, the European Parliament introduced the Renewable Energy Directive (2009/28/EC), which is the legal framework for the development of renewable energy across all sectors of the EU economy. Since then, the deployment of renewables has kept growing yearly, reaching more than 22% in 2020 [9]. In December 2018, the revised Renewable Energy Directive (Directive (EU) 2018/2001) entered into force and it established a new binding renewable energy target for the EU for 2030 of at least 32% of final energy consumption, with a clause for a possible upwards revision by 2023 and an increased 14% target for the share of renewable fuels in transport by 2030. Nowadays, the most-used alternatives to fossil fuels is biomass (~24% of the overall renewable supply of energy and over 95% of renewal heat production in Europe [10,11]) to produce heat and power, because biomass is considered a carbon-neutral fuel due to its renewability [12,13], and it can contribute to reducing the emission of greenhouse gases. It is predicted that, by 2023, the amount of biomass used for energy production could double [14]. The process of biomass combustion generates huge amounts of biomass ash. Currently, approximately 50 Mt/yr of ash from biomass combustion is generated in Europe [10]. In 2019, in Belgium, renewable energies made up 8% of the total energy supply and approximately 10% of the final energy consumption, and approximately 64% of that renewable energy came from biomass [15,16]. In Wallonia, the situation is similar, where biomass is the main source of renewable energy and accounts for a bit more than 40% of the total production of renewable energy (Figure 1).

The production of BA has been increasing every year in Europe, reaching 5.5 million tons in 2020 [17,18]. Combustion or co-combustion of biomass with coal can reduce coal consumption and minimize global $CO_2$ emissions. Biomass accounts for more than 4% of the total energy consumption in the European Union and will increase in the future. However, BA does not currently have any established valorization process and its storage occupies large areas and increases the risk of groundwater contamination. Currently, BA is only commonly used as a soil supplement to improve alkalinity in agricultural applications [19]. Recent research results have shown the suitability of BA as a partial replacement for cement in structural concrete for buildings [20–26]. Esteves [27] showed that the total chemical composition of silica, alumina and ferric compounds could range from 18.6% to 59.3% for the wood ash samples examined. Wang [28] reported that wood waste fly ash consisted of very irregularly shaped particles with a higher porous surface area than FA. The mechanisms of interaction with cement and the effect of undesirables, especially alkali and phosphorus, remain to be explored.

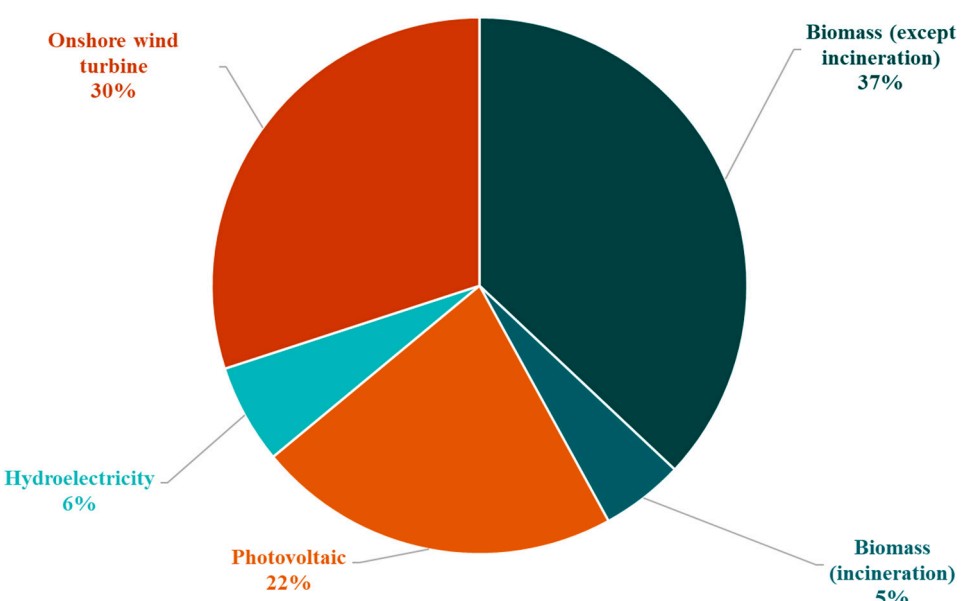

**Figure 1.** Biomass share in the total production of renewable energy in Wallonia (Belgium).

An interesting application of BA is considered for mortars used in the construction of buildings. Masonry work is carried out by combining bricks with a mortar that acts as a bonding agent [29]. Masonry mortar is defined as a paste obtained by mixing [30] water, a mixture of fine aggregates and a binder: clay, gypsum, lime, cement or a combination of several of these. The resulting material is capable of setting and hardening [30]. Masonry mortar is one of the most widely used building materials, specifically in Northwestern Europe. The use of mortar in construction work is considered as the foundation of ancient civilization: mixtures were simply prepared using the combination of lime, natural pozzolan, sand and water. Moreover, recycled materials were used as a substitute for sand: recycled crushed brick, residue from cutting marble slabs, etc. Mortar-based lime was replaced by the invention of Portland cement in the 19th century. This replacement was mainly due to the improved mortar properties that Portland cement could provide [29]. Properties masonry mortars have to fulfill are:

-   aesthetically acceptable and beautiful;
-   protect the interior structure from the penetration of air, water or chemical elements into a masonry assembly;
-   help support the weight placed on the wall;
-   seal the joints to provide a weather-resistant wall.

Good mortar is vitally important in any brick or block wall as it binds the units together.

A rendering is a layer of mortar applied to a wall or, on the outside, to the insulating mantle of this wall. This coating can also be found in the form of roughcast, which is a coating applied to a wall. Rendering mortars are generally of mineral origin, such as cement mortar, lime mortar or plaster (gypsum). They are composed of a binder (lime, plaster, Portland cement or clay) and mineral fillers (aggregates). The addition of colored pigments is not essential and varies according to the desired effect. The rendering mortar also has a number of objectives to fulfil:

-   mechanically resistant;
-   aesthetically pleasing, as it is visible on part of the wall;
-   waterproof but permeable to water vapor.

A great deal of work has been carried out by the construction products industry to develop and market new premixed products for the preparation of plastering mortars.

Unfortunately, the results obtained have not always proved durable over time. The main causes of rapid pathological degradation of this type of coating are [31,32]:

- incompatibility between the base material and the mortar;
- poor plaster layering;
- poor knowledge of mortar application techniques.

However, cementitious materials with BA lack building codes and data on their long-term behavior. In this research, the substitution limits of BA are studied by comparison of the mechanical strengths, fresh state properties and hardened properties of mortars produced with BA and FA (stage 1). The viability of the process is then tested against the requirements for two mortar applications [33]: masonry and rendering mortars (stage 2). These applications represent an original but possible market for expanding the production of FA. Durability factor such as porosity is investigated with regards to substation rate and age.

## 2. Comparison of Fly Ash with Biomass Ash

### 2.1. Physico-Chemical Characterization of Fly Ash

Two types of fly ash are used: the FA comes from a historical deposit in the Liege area and the BA was produced in the thermal power plant of Awirs based on 100% wood resources (spruce) in the form of pellets. The main physico-chemical characteristics of both ashes are given in Tables 1 and 2.

**Table 1.** Physical properties of fly ashes.

| | $D_{10}$ (µm) | $D_{50}$ (µm) | $D_{90}$ (µm) | Specific Surface ($m^2$/g) BET ($m^2$/g) | Specific Surface ($m^2$/g) (Laser Granulometry) ($m^2$/g) | Specific Surface Blaine ($m^2$/g) | Density |
|---|---|---|---|---|---|---|---|
| FA | 4.63 | 29.04 | 88.65 | 2.1790 | 0.68 | 0.5582 | 2.75 |
| BA | 5.01 | 33.39 | 114.31 | 2.6119 | 0.63 | 0.4984 | 2.61 |

**Table 2.** Chemical characteristics (main oxides) of fly ashes and cement by XRF (fused disks).

| | $SiO_2$ | $Al_2O_3$ | $Fe_2O_3$ | MnO | MgO | CaO | $Na_2O$ | $K_2O$ | $P_2O_5$ | $TiO_2$ | LOI |
|---|---|---|---|---|---|---|---|---|---|---|---|
| FA (%) | 49.3 | 27.7 | 7.9 | 0.1 | 1.6 | 1.4 | 0.8 | 4 | 0.3 | 1 | 5.9 |
| BA (%) | 24.7 | 5.3 | 3.2 | 1 | 9.3 | 25.8 | 2.3 | 7.9 | 4.9 | 0.4 | 9.7 |
| CEM I 52.5 N | 20.5 | 4.8 | 3.4 | - | - | 63.6 | 0.83 | - | - | - | 1.5 |

The densities of fly ash (FA and BA) are lower than those of cement. The specific surface areas determined by the Blaine method (in accordance with standard EN 196-6) and by the BET method using $N_2$ are also presented in this table: the specific surface area of materials measured by the BET method is two to five times greater than that measured by the Blaine method, which corresponds to materials with high porosity such as fly ashes [2].

BA is characterized by a lower [$SiO_2$ + $Al_2O_3$ + $Fe_2O_3$] content than FA (33.2 and 84.9%, respectively); this should result in lower pozzolanic reactivity. BA has a higher loss on ignition than FA, which may be due to incomplete combustion of carbon due to kinetics and mass transfer in the biomass plant, when the wood pellets are at a temperature between 750 °C and 1000 °C. In fact, some of them may not be burnt properly.

Alkaline materials have to be limited in cementitious mixes as they can induce an alkali aggregate reaction (AAR) when in contact with aggregates containing amorphous silicium oxides: chemical reaction may occur in specific conditions (temperature and humidity), leading to gel production and risk of cracking [34]. Moreover, it is observed that the temperature increased rapidly at the beginning of mixing BA with the solution containing alkali; this is a result of a highly exothermic reaction of $P_2O_5$. The BA phosphate

soluble in water elements may react with the calcium or sodium oxides and release a lot of heat [35]. A similar heat release trend is observed in the hydration of magnesium phosphate cement [36].

### 2.2. Mortars Design and Testing

To assess the influence of the partial replacement of Portland cement by BA and to compare it with the influence of the FA, a series of mortar mixtures are prepared and characterized, as shown in Figure 2.

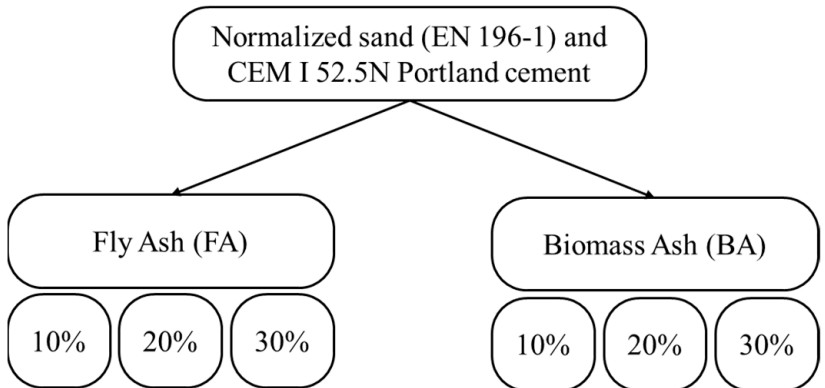

**Figure 2.** Stage 1 sample production.

A reference mortar containing water, CEM I 52.5 N cement (ordinary Portland cement) and standardized sand is prepared according to EN 196-1. Then, part of the cement is substituted with FA or BA at substitution rates of 10%, 20% and 30% (in volume).

The exact composition of mortar mixtures is given in Table 3. Water to binder (W/B) ratio remains constant and equal to 0.5 for all the mortars tested at this stage.

**Table 3.** Composition of mortar mixtures.

| Mortar / Materials | Ref | FA | | | BA | | |
|---|---|---|---|---|---|---|---|
| | | 10% | 20% | 30% | 10% | 20% | 30% |
| Standard Sand [kg/m$^3$] | 1350 | | 1350 | | | 1350 | |
| CEM I 52.5 N [kg/m$^3$] | 450 | 405 | 360 | 315 | 405 | 360 | 315 |
| FA [kg/m$^3$] | 0 | 45 | 90 | 135 | | 0 | |
| BA [kg/m$^3$] | 0 | | 0 | | 45 | 90 | 135 |
| Water [kg/m$^3$] | 225 | | 225 | | | 225 | |

The test performed on the different mixes is related to workability (EN 1015-3), density (EN 1015-6), mechanical strength (EN 196-1), apparent porosity and water absorption by immersion (NBN B15-215). The objective is to determine the maximum acceptable substitution rate of Portland cement by BA. Three samples at least are prepared for each test (six for compressive strength). Tests at fresh state are performed directly after mixing.

### 2.3. Results of the Sensitivity Study on The Substitution Rate

While FA induces a slight decrease in workability, BA dramatically impacts the fresh behavior of mortars (Figure 3). A higher specific surface usually explains this type of behavior [23,37]: when BA content increases, adsorption increases and the water available for workability decreases.

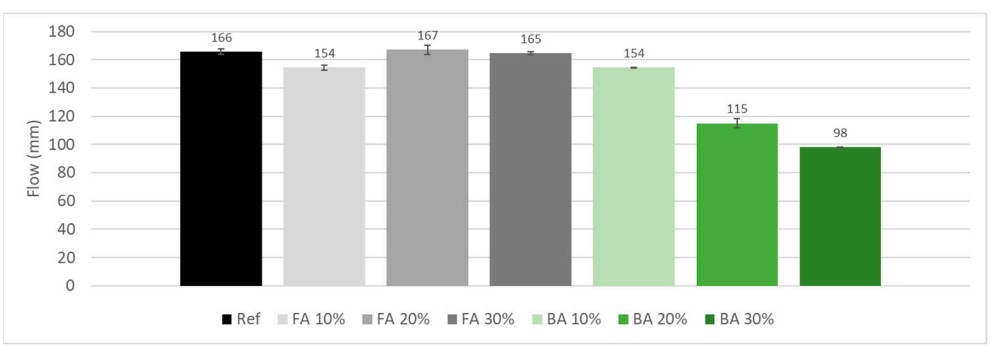

**Figure 3.** Spreading of fresh mortars for the different mixtures considered.

Density of hardened mortars (Figure 4) increases with time. The FA substitution does not seem to impact the density of mortar and it remains unchanged for substitution rates up to 20% of BA. For 30% BA, a lot of air bubbles were present in the mix and induced a large decrease in samples' mass. This phenomenon was observed previously [38] and is attributed to coal particle residues which are able to modify the surface free energy of water and favor air entrapment in liquid mixes.

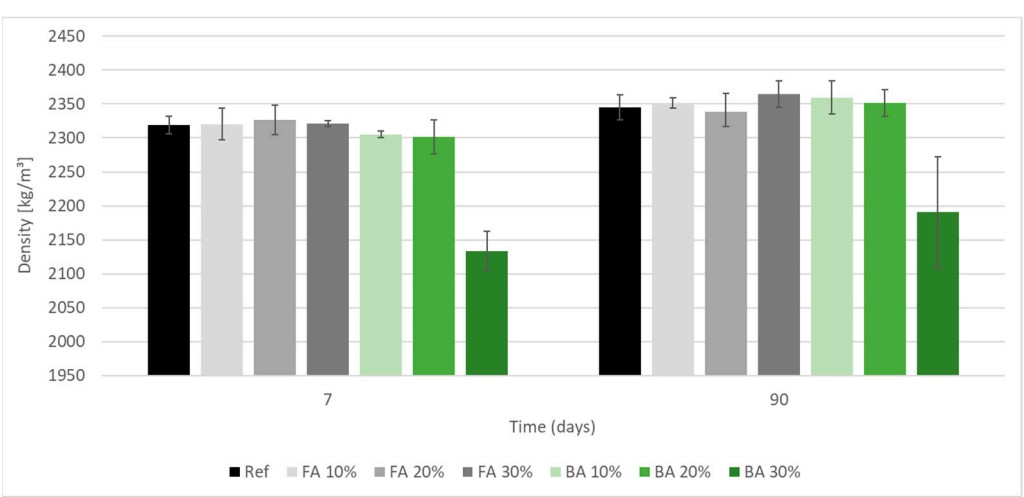

**Figure 4.** Density of hardened mortars 7 days and 90 days after casting.

Mechanical performances clearly show a positive evolution with time, for both types of fly ash (Figure 5). The increase is, however, higher with FA than with BA, which attests to a higher pozzolanic activity. BA seems, however, to have a filler effect which contributes to diminish detrimental effect of substitution.

The important decrease in mechanical performances for 30% BA means that the substitution rate should be limited to a value around or lower to 20% for which the compressive strength loss remains acceptable (25% for BA vs. 10% for FA cf. Table 4) and the absolute value of the compressive strength remains high (48 MPa) with a low coefficient of variation.

Water absorption by immersion after 28 days tests are conducted according to NBN B15-215 and three samples are tested for each composition. As can be seen in Figure 6, the water absorption level reached are reasonably low for all the mixtures, but a clear increase can be observed with increasing substitution rates. BA mortars also exhibit higher water absorption and porosity than mortar prepared with classical FA (Table 5).

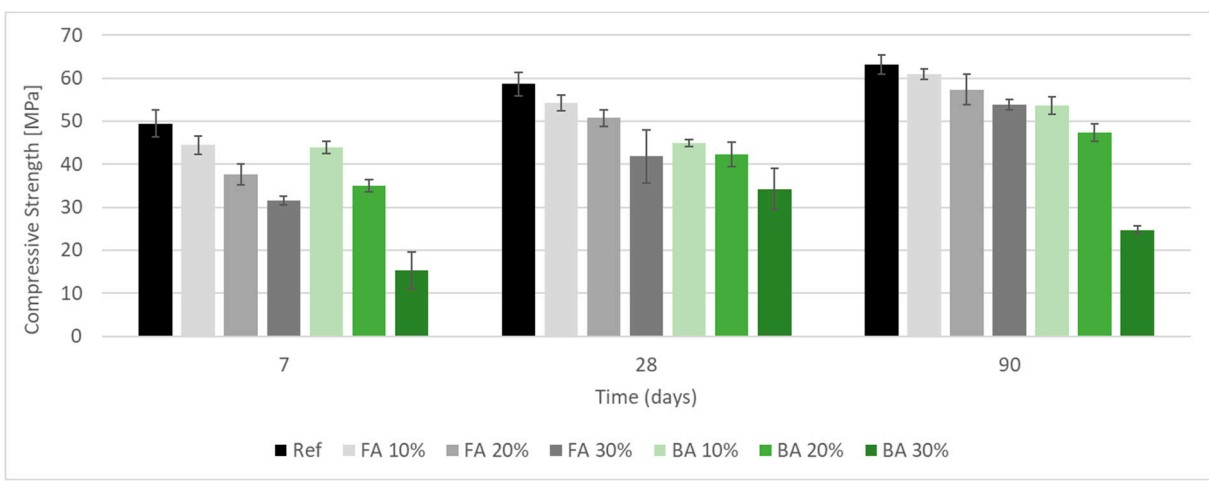

**Figure 5.** Evolution of the compressive strength with time for all the compositions.

**Table 4.** Loss of compressive strength compared with the reference mortar.

| | Loss of Compressive Strength Compared to the Reference Mortar (%) | | | | | |
|---|---|---|---|---|---|---|
| | **FA** | | | **BA** | | |
| | **10%** | **20%** | **30%** | **10%** | **20%** | **30%** |
| 7 days | 10 | 24 | 36 | 11 | 29 | 69 |
| 28 days | 4 | 9 | 15 | 23 | 28 | 42 |
| 90 days | 8 | 14 | 29 | 15 | 25 | 61 |

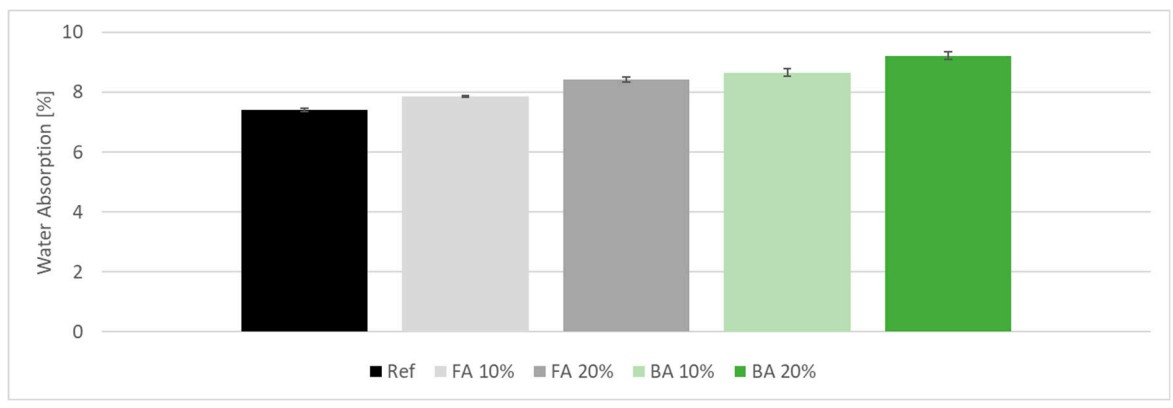

**Figure 6.** Water absorption by immersion for the different composition tested.

**Table 5.** Water absorption and porosity of the different mortar mixtures.

| | | Water Absorption (% in Mass) | Porosity (% in Volume) |
|---|---|---|---|
| | Ref | 7.4 | 15.9 |
| FA | 10% | 7.9 | 16.6 |
| | 20% | 8.4 | 17.5 |
| | 30% | - | - |
| BA | 10% | 8.7 | 17.8 |
| | 20% | 9.2 | 19.1 |
| | 30% | - | - |

### 3. Development of Masonry and Rendering Mortars with Biomass Ash

The investigation of the viability of masonry and rendering mortars produced with BA is now proposed. Based on the previous sensitivity study, the substitution rate is kept under 20%. Rendering mortars must present higher workability than masonry mortar [37] and thus have lower cement quantity (246 kg/m$^3$ vs. 369 kg/m$^3$) and higher W/B ratio (1.02 vs. 0.65). The mixtures have been designed to obtain a constant spreading flow diameter of 175 ± 10 mm by slightly adapting the W/B ratio. The different samples produced for this second stage are shown in Figure 7.

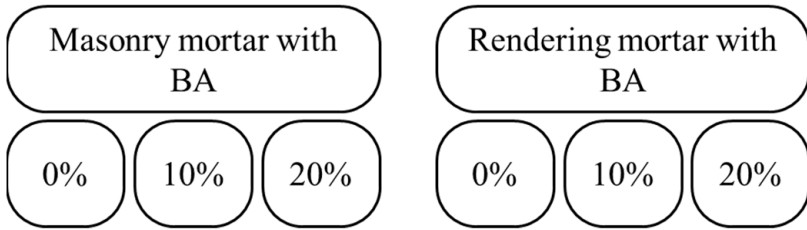

**Figure 7.** Stage 2 sample production.

The samples were produced using water, a 0/2 mm yellow siliceous Rhine sand and CEM II/B-M (S-V) 32.5 N cement. This cement, which is less reactive than CEM I but modified with 18 to 30% of slags and fly ashes, is particularly suitable for mortars. Moreover, 32.5 strength class is largely enough for such applications. The exact composition of the different mortar mixtures is available in Table 6.

**Table 6.** Composition of the masonry and rendering mortars developed.

| Mortar<br>Materials | Masonry | | | Rendering | | |
|---|---|---|---|---|---|---|
| Substitution Rate [%] | 0% | 10% | 20% | 0% | 10% | 20% |
| Yellow Rhine Sand [kg/m$^3$] | | 1350 | | | 1350 | |
| CEM III 32.5 N [kg/m$^3$] | 369 | 332 | 295 | 246 | 222 | 197 |
| BA [kg/m$^3$] | 0 | 35 | 70 | 0 | 23 | 47 |
| Water [kg/m$^3$] | 240 | 257 | 256 | 251 | 250 | 249 |
| W/B | 0.65 | 0.7 | 0.7 | 1.02 | 1.02 | 1.02 |

Mechanical performances of both mortars decrease with an increasing substitution rate (Figure 8), and the loss of compressive strength compared with the reference mortars are visible in Table 7. The decrease can reach 27% but the absolute values remain more than satisfying for these applications. The level of compressive strength is lower than for standardized reference mortars (Figure 5), which is normal for this type of mortar (lower resistance and higher deformability). Masonry mortars are also more resistant than rendering mortars. However, M5 and M10 mortars (according to EN 998-2) are the most used for masonries: they correspond to compressive strength at 28 days of 5 and 10 MPa, respectively. Results obtained with biomass ashes are largely enough for such applications. Moreover, the compressive strength of mortars with biomass ash increases with time, which attests to the residual chemical activity of pozzolanic components and not only the filler effect.

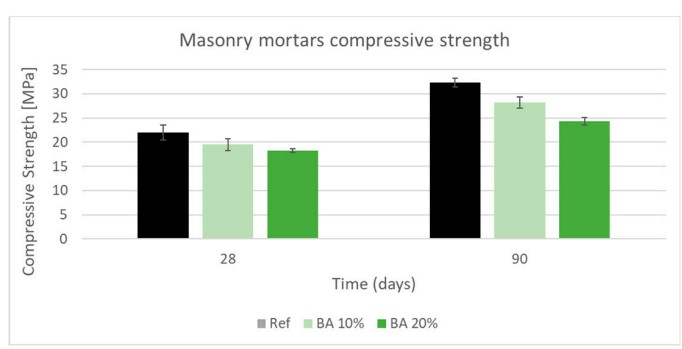
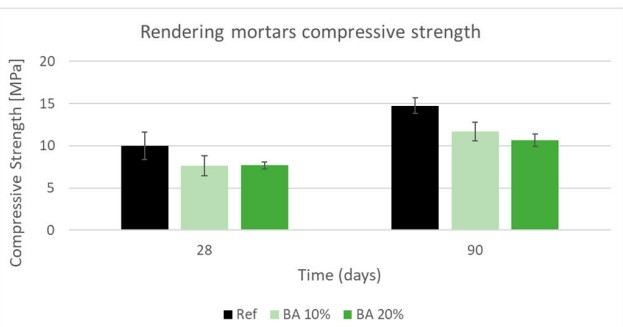

**Figure 8.** Compressive strength of the masonry (**left**) and rendering (**right**) mortars for the different BA substitution rates.

**Table 7.** Loss of compressive strength with increasing substitution rates for both masonry and rendering mortars.

| | Loss of Compressive Strength Compared with Reference Mortars (%) | | | |
|---|---|---|---|---|
| | **Masonry Mortar** | | **Rendering Mortar** | |
| | **10%** | **20%** | **10%** | **20%** |
| 28 days | 11 | 17 | 23 | 23 |
| 90 days | 13 | 25 | 20 | 27 |

The loss of resistance (Table 7) seems to be higher for rendering mortars, which is due to higher water to binder ratio.

The porosity (Table 8) and water absorption by immersion (Figure 9) of masonry mortars and plastering mortars are relatively equivalent. They remain relatively low with regard to reference mortars which confirms that biomass ashes may be used for such applications, even with a 20% substitution rate.

**Table 8.** Porosity and water absorption by immersion for masonry and rendering mortar with variable BA substitution rates.

| | | Water Absorption (% in Mass) | Porosity (% in Volume) |
|---|---|---|---|
| | 0% | 10.5 | 21.3 |
| Masonry | 10% | 11.5 | 23 |
| | 20% | 12.3 | 24.1 |
| | 0% | 12.2 | 24.1 |
| Rendering | 10% | 11.2 | 22.4 |
| | 20% | 10.5 | 23.7 |

Masonry mortars have a higher B/S ratio than rendering mortar and therefore have a greater volume of binder, explaining their higher mechanical strength, lower porosity and lower water absorption. For the other mixes (10 and 20% substitution rate), the difference between the two B/S ratios has no influence on the values of apparent porosity and water absorption by immersion.

As it can be seen in Figure 9, the water absorption of the masonry mortar increases with the substitution rate, which is in agreement with the literature [23], while the rendering mortar does not seem to be impacted.

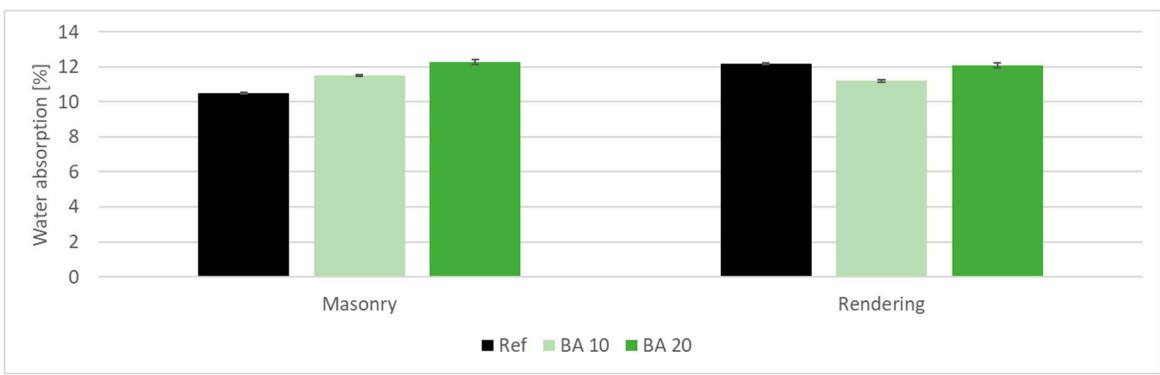

**Figure 9.** Water absorption by immersion of the masonry (left) and rendering (right) mortars for the different BA substitution rates.

### 4. Conclusions

The following conclusions may be drawn from the present investigations concerning the suitability of biomass fly ashes for use in cement-based composites:

- The mineralogy of FA and BA is different, inducing a lower reactivity of BA;
- The workability of fresh mortar is much more affected by the addition of BA than by the addition of FA, which is probably due to the rougher shape of BA;
- Density does not seem to be affected by the addition of FA or of BA, except for the BA 30% mortar for which air bubbles appeared and led to a significant drop in density;
- The mechanical strength of mortars deteriorates more for mortars containing BA than for mortars containing FA, even if it remains higher than 30 MPa for a substitution rate of 20% at 7 days;
- When FA is added to the mortar mix, the long-term mechanical performance (90 days) is generally improved. However, above a 30% substitution, the long-term mechanical performance starts to decrease;
- BA induces a higher water absorption by immersion and porosity than FA.

This first stage clearly show that BA may be used for manufacturing masonry and rendering mortars up to a 20% substitution rate. Two types of mixes were designed and tested:

- BA does not affect the workability of these mortars as the water demand remains almost identical for all substitution rates;
- Density does not seem to be affected by the addition of BA;
- The mechanical strength of mortars is affected by the addition of BA with a significant loss of compressive strength of up to 27% for rendering mortar with 20% BA. Mechanical performances remain, however, higher than the required value for masonry or rendering mortars.

Based on the tests conducted in this study, it can be concluded that, even though BA has an impact on mortar properties, it can be used in cementitious materials as a substitute for Portland cement at a low level of replacement (between 5 and 15%).

These results are, of course, only valuable for the BA tested in this study and increasing the substitution rate could be a possibility with BA produced through a well-defined, better-controlled combustion process, followed by appropriate treatment. Nevertheless, further experiments on durability should be carried out before providing a final recommendation.

**Author Contributions:** Conceptualization, L.C. and J.H.; methodology, L.C. and J.H.; validation, S.G., F.M. and Z.Z.; formal analysis, L.C.; investigation, J.H. and Z.Z.; resources, J.H. and Z.Z.; data curation, F.M.; writing—original draft preparation, L.C.; writing—review and editing, J.H.; visualization, L.C.; supervision, J.H.; project administration, L.C.; funding acquisition, L.C. All authors have read and agreed to the published version of the manuscript.

**Funding:** Regional Government of Wallonia (Belgium) and the European Regional Development Fund through ECOLISER (eco-binders for soil treatment, waterproofing mem-branes and roads) research project (2016–2020).

**Data Availability Statement:** The data presented in this study are available on request from the corresponding author.

**Acknowledgments:** The authors would like to acknowledge the Regional Government of Wallonia (Belgium) and the European Regional Development Fund for their financial support through ECOLISER (eco-binders for soil treatment, waterproofing membranes and roads) research project (2016–2020). Authors warmly thank Marine Meys, a Master's student, who performed most of the tests.

**Conflicts of Interest:** The authors declare no conflict of interest.

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
