# Peer review of "Development and Properties of Recycled Biomass Fly Ashes Modified Mortars"

_recycling, doi:10.3390/recycling9030046_

Round 1
Reviewer 1 Report
Comments and Suggestions for Authors
1. The abstract needs revision. Stick to commonly used terminologies. For instance, clarify "classical fly ash" as ASTM C618 compliant fly ash. Specify "mechanical strength" as compressive strength. Avoid mixing terms like "biomass fly ash"; it's either biomass or fly ash. The abstract's findings are too general. Specify the properties that make them suitable.
2. Avoid introducing new terms like "CFA"; stick to "FA" throughout the manuscript.
3. Line 37: Correct the statement about FA producing CH. Define "masonry cement" and its distinction from conventional C150 cements. Note that FA reacts with any alkaline solution, not just cement.
4. Line 40: Replace "good quality" FA with ASTM compliant FA and refer to "bad quality" FA as off-spec FA, common terms in literature.
5. Line 41: Change "Biomass fly ash" to biomass ash or BA.
6. Line 43: Distinguish between biochar and incinerated BA in the presence of oxygen.
7. Line 46: Clarify "renewables" to possibly mean renewable energy.
8. Explain the significance of the Belgian case study and its representativeness for Europe.
9. Consider removing repetitive content in lines 53-60.
10. Strengthen the introduction with a more comprehensive literature review, especially regarding biomass use in concrete.
11. Clarify the purpose and objectives of the research in lines 67-72 to align with earlier sections.
12. Include a comparison of EN 197 and ASTM C618 for readers outside Europe. Refer to Suraneni et al. (2021) for more information. Suraneni, Prannoy, et al. "ASTM C618 fly ash specification: Comparison with other specifications, shortcomings, and solutions." ACI Mater. J 118.10.14359 (2021): 51725994.
13. Restructure section 2 to align with standard Materials and Methods sections. Move information about EN196 to the introduction.
14. Ensure consistency in terminology, avoiding terms like "good quality" or "well-accepted" FA.
15. Include cement oxide composition in Table 2.
16. Discuss the tests performed to enhance the scientific credibility of the paper.
17. Clarify if section 2.3 is part of the results section or methods section to avoid mixing them.
18. Elaborate on results, explaining figures, patterns observed, and comparisons with existing literature.
19. Move the description of experiments in the first part of section 3 to the Materials and Methods section for clarity and standardization.
Comments on the Quality of English LanguageThe choice of certain terminologies in the manuscript appears to be incorrect. It is recommended that the authors use terms that are conventional within this domain for better clarity and understanding.
Author Response
Reviewer 1
- The abstract needs revision. Stick to commonly used terminologies. For instance, clarify "classical fly ash" as ASTM C618 compliant fly ash. Specify "mechanical strength" as compressive strength. Avoid mixing terms like "biomass fly ash"; it's either biomass or fly ash. The abstract's findings are too general. Specify the properties that make them suitable.
The abstract has been reviewed and completed in terms of resuslts and findings (with repcet to maximum number of words).
- Avoid introducing new terms like "CFA"; stick to "FA" throughout the manuscript.
Clarification about fly ashes has been implemented in the paper (BA for biomass ash and FA for compliant fly ashes. Figures and tables have been adapted
- Line 37: Correct the statement about FA producing CH. Define "masonry cement" and its distinction from conventional C150 cements. Note that FA reacts with any alkaline solution, not just cement.
Correction has been proposed. The two types of mortars are more deeply presented in the introduction
- Line 40: Replace "good quality" FA with ASTM compliant FA and refer to "bad quality" FA as off-spec FA, common terms in literature.
- Done
- Line 41: Change "Biomass fly ash" to biomass ash or BA.
See above
- Line 43: Distinguish between biochar and incinerated BA in the presence of oxygen.
- Done. Pyrolysis has been removed in order to avoid confusion
- Line 46: Clarify "renewables" to possibly mean renewable energy.
Deep explanation and presentation of renewable energies is proposed in the texte as well as a description of the European situation
- Explain the significance of the Belgian case study and its representativeness for Europe.
See just above
- Consider removing repetitive content in lines 53-60.
- Introduction has been adapted and completly rewritten
- Strengthen the introduction with a more comprehensive literature review, especially regarding biomass use in concrete.
See just above
- Clarify the purpose and objectives of the research in lines 67-72 to align with earlier sections.
- Done
- Include a comparison of EN 197 and ASTM C618 for readers outside Europe. Refer to Suraneni et al. (2021) for more information. Suraneni, Prannoy, et al. "ASTM C618 fly ash specification: Comparison with other specifications, shortcomings, and solutions." ACI Mater. J 118.10.14359 (2021): 51725994.
The authors would like to thank the reviewer for the recommendation and the reference of the publication, which is indeed very interesting. We believe, however, that the values given in the text are sufficiently clear for us to fully understand the requirements for this type of product.
- Restructure section 2 to align with standard Materials and Methods sections. Move information about EN196 to the introduction.
We moved the information about EN to introduction. However, we feel the organization in stage and stage 2 more clear for the reader and we kept the structure as it is.
- Ensure consistency in terminology, avoiding terms like "good quality" or "well-accepted" FA.
See above
- Include cement oxide composition in Table 2.
- Done
- Discuss the tests performed to enhance the scientific credibility of the paper.
Each test is discussed and often compared to litterature
- Clarify if section 2.3 is part of the results section or methods section to avoid mixing them.
See above
- Elaborate on results, explaining figures, patterns observed, and comparisons with existing literature.
Tests have been analysed more deeply and conclusions have been partially renewed
- Move the description of experiments in the first part of section 3 to the Materials and Methods section for clarity and standardization.
See above
Comments on the Quality of English Language
The choice of certain terminologies in the manuscript appears to be incorrect. It is recommended that the authors use terms that are conventional within this domain for better clarity and understanding.
We have completely revised the text and incorporated the terminology found in the standards and recommended by the reviewer, so that we have a clear and precise text.
Reviewer 2
In my opinion, this article does not present novelty. In recent years, numerous articles on biomass fly ash for its application in mortars have been studied and published.
Bibliography has been expanded
- First of all, it should be noted that the introduction does not clearly include the objectives of this work, so the Authors should provide more content in the introduction section.
The introduction has been improved in terms of content and form: aspects relating to types of mortar, the production of biomass ash with regard to the production of renewable energy and the use of ash in concrete have been explained in detail to achieve the aim of the publication.
- Secondly, images related to the biomass fly ash used could be included and the origin explained more broadly, that is, what type of biomass is burned in said plant.
Based on 100% wood resources (spruce) in the form of pellets. Text has been modified
- Third, the dosages are not carried out appropriately, since cement has a higher density than coal bottom ash and biomass fly ash, so adjustments must be made in the composition of the mortars.
Substitution rates are given in volume in order to take it into account (line 171 new draft)
- The property of the cement must be included. It is used in a specific table that describes the typology of cement in the same table and not only in a paragraph
Done in Table 2
- The results are presented in figures with bars, but in my opinion a table with the results should be included to be able to assess the content of the results quantitatively.
We fully agree. We however use not to duplicate the results. Values are of course available for any researcher who would be interested. Moreover, the objective of this paper was a comparison between BA and FA and that’s the reason why we preferred to have table incorporating « loss of » .
- No correlations have been established between variables, which I consider important and which emerge from the results obtained.
Tests have been analysed more deeply and conclusions have been partially renewed.

Reviewer 2 Report
Comments and Suggestions for Authors
In my opinion, this article does not present novelty. In recent years, numerous articles on biomass fly ash for its application in mortars have been studied and published.
- First of all, it should be noted that the introduction does not clearly include the objectives of this work, so the Authors should provide more content in the introduction section.
- Secondly, images related to the biomass fly ash used could be included and the origin explained more broadly, that is, what type of biomass is burned in said plant.
- Third, the dosages are not carried out appropriately, since cement has a higher density than coal bottom ash and biomass fly ash, so adjustments must be made in the composition of the mortars.
- The property of the cement must be included. It is used in a specific table that describes the typology of cement in the same table and not only in a paragraph
- The results are presented in figures with bars, but in my opinion a table with the results should be included to be able to assess the content of the results quantitatively.
- No correlations have been established between variables, which I consider important and which emerge from the results obtained.
Author Response
Reviewer 2
In my opinion, this article does not present novelty. In recent years, numerous articles on biomass fly ash for its application in mortars have been studied and published.
Bibliography has been expanded
- First of all, it should be noted that the introduction does not clearly include the objectives of this work, so the Authors should provide more content in the introduction section.
The introduction has been improved in terms of content and form: aspects relating to types of mortar, the production of biomass ash with regard to the production of renewable energy and the use of ash in concrete have been explained in detail to achieve the aim of the publication.
- Secondly, images related to the biomass fly ash used could be included and the origin explained more broadly, that is, what type of biomass is burned in said plant.
Based on 100% wood resources (spruce) in the form of pellets. Text has been modified
- Third, the dosages are not carried out appropriately, since cement has a higher density than coal bottom ash and biomass fly ash, so adjustments must be made in the composition of the mortars.
Substitution rates are given in volume in order to take it into account (line 171 new draft)
- The property of the cement must be included. It is used in a specific table that describes the typology of cement in the same table and not only in a paragraph
Done in Table 2
- The results are presented in figures with bars, but in my opinion a table with the results should be included to be able to assess the content of the results quantitatively.
We fully agree. We however use not to duplicate the results. Values are of course available for any researcher who would be interested. Moreover, the objective of this paper was a comparison between BA and FA and that’s the reason why we preferred to have table incorporating « loss of » .
- No correlations have been established between variables, which I consider important and which emerge from the results obtained.
Tests have been analysed more deeply and conclusions have been partially renewed.

Round 2
Reviewer 1 Report
Comments and Suggestions for Authors
The authors left the 'Track Changes' feature enabled, making it difficult to follow the modifications and corrections. Please resubmit the manuscript with all changes accepted and highlighted.
Comments on the Quality of English LanguageI will provide comments after the correct version is submitted.
Author Response
Thanks for pointing it out, please find the highlighted version.
Reviewer 2 Report
Comments and Suggestions for Authors
The authors have incorporated new results and have modified the questions raised by the reviewers, so in my opinion this article can be accepted for publication.
Author Response
Thank you.
Round 3
Reviewer 1 Report
Comments and Suggestions for Authors
I must express that there has not been significant improvement in the organization of the paper. The Methods and Results sections remain intertwined, which could lead to confusion and detract from the clarity of the presentation. Furthermore, the manuscript continues to be filled with numerous grammatical errors. A comprehensive proofreading is essential to address these issues and elevate the quality of the paper to a scholarly standard.
Below are more detailed comments:
1. Line 19: "Water absorption remains however more or less equivalent which means good durability." This is a misleading statement. How much was the water absorption? Both of the ashes could lead to high water absorption which may not indicate good durability. Instead, please be more specific and state the values.
2. Line 31: "interesting" is not a good choice of word here.
3. Line 44: Fly ash and bauxite have different chemical compositions. Please elaborate more on how fly ash is used as a substitute for bauxite.
4. Line 74: You already introduced BA. No need to put it in parentheses again. Please check your text when using AI to avoid typos like this!
5. Line 96: There is a clear disconnect between previous paragraphs and the paragraphs starting from line 96. This part seems to be copied from ChatGPT and does not add real value. The authors are attempting to justify the novelty of their work by adding this part, but after reading it, I am not sure why they wrote 3 pages on the state of renewable energy and suddenly shift gears to mortar and rendering. If the focus of the study is on mortar and rendering, I suggest revising the introduction and cutting to the chase without beating around the bush.
6. Please add the units in Table 1 and do not rotate the headings.
7. What is the SO3 content of FA and BA?
8. BA has a high K2O and P2O5 content as expected from any bioash. Some explanations regarding the potential impacts of K2O and P2O5 on the hydration of cement are warranted.
9. Line 152: Porosity or amorphousness?
10. What does line 154 mean?
11. Was the XRF performed on pressed pellets or fused disks? The technical information of all tests should be provided before the results are presented.
12. So, if SiO2 + Al2O3 + Fe2O3 of BA is 33.2%, how do you justify its application based on ASTM C618 since it doesn’t pass the minimum 50% requirement?
13. Line 159: How is the combustion temperature of wood different from coal? I think wood burns much easier than coal, so why is it that it has a higher LOI? There should be other reasons than what is currently provided in the text that contributes to the higher LOI of wood.
14. The density of BA is lower than FA so, so the lower density of BA means that for the same weight, BA will occupy a larger volume than FA. This is critical in mixtures like mortar where the distribution and packing of particles can significantly affect the properties of the finished material. A detailed justification and explanation of the choice of replacement levels, considering its impact on both the chemical and physical properties of the mortar, is necessary.
15. Line 175: The authors provided a superficial explanation regarding the tests performed. Please expand this section and clearly explain:
· how many samples were used for each test?
· what were the shape and size of the samples?
· how the samples were cured?
· at what age were the samples tested?
16. Section 2.2 title says mortar design and testing, but you are including the results of the tests. Then, in section 2.3, you have masonry rendering and mortars and show similar tests and results. The whole structure of section 2 is very confusing and must be revised completely.
17. I do not see much discussion on the results. The authors simply explained the figures and did not explain what factors contributed to those results. How do your results compare with the existing literature? What new findings did you discover?
Comments on the Quality of English LanguageA comprehensive proofreading is essential to address these issues and elevate the quality of the paper to a scholarly standard.
Round 4
Reviewer 1 Report
Comments and Suggestions for Authors
Thank you for your efforts in revising your manuscript. However, after three rounds of reviews, it is clear that many of the key issues, including the necessary separation of methods and results, have not been adequately addressed. Furthermore, the responses to the comments continue to lack depth. Therefore, I regret to t reject this submission.
Comments on the Quality of English LanguageThe text still contains many typos. "alcaline" in line 169 should be changed to alkaline.
Author Response
Thanks. Last corrections have been produced.